# Accounting for heading date gene effects allows detection of small-effect QTL associated with resistance to Septoria nodorum blotch in wheat

**Luis A. Rivera-Burgos**[1], **Gina Brown-Guedira**[2], **Jerry Johnson**[3], **Mohamed Mergoum**[3], **Christina Cowger**[2]*

**1** Department of Crop and Soil Sciences, North Carolina State University, Raleigh, North Carolina, United States of America, **2** U.S. Department of Agriculture – Agricultural Research Service, North Carolina State University, Raleigh, North Carolina, United States of America, **3** Department of Crop and Soil Sciences, University of Georgia, Athens, Georgia, United States of America

* ccowger@ncsu.edu, Christina.Cowger@usda.gov

## Abstract

In humid and temperate areas, Septoria nodorum blotch (SNB) is a major fungal disease of common wheat (*Triticum aestivum* L.) in which grain yield is reduced when the pathogen, *Parastagonospora nodorum*, infects leaves and glumes during grain filling. Foliar SNB susceptibility may be associated with sensitivity to *P. nodorum* necrotrophic effectors (NEs). Both foliar and glume susceptibility are quantitative, and the underlying genetics are not understood in detail. We genetically mapped resistance quantitative trait loci (QTL) to leaf and glume blotch using a double haploid (DH) population derived from the cross between the moderately susceptible cultivar AGS2033 and the resistant breeding line GA03185-12LE29. The population was evaluated for SNB resistance in the field in four successive years (2018–2021). We identified major heading date (HD) and plant height (PH) variants on chromosomes 2A and 2D, co-located with SNB escape mechanisms. Five QTL with small effects associated with adult plant resistance to SNB leaf and glume blotch were detected on 1A, 1B, and 6B linkage groups. These QTL explained a relatively small proportion of the total phenotypic variation, ranging from 5.6 to 11.8%. The small-effect QTL detected in this study did not overlap with QTL associated with morphological and developmental traits, and thus are sources of resistance to SNB.

## Introduction

Septoria nodorum blotch (SNB), caused by the fungal pathogen *Parastagonospora nodorum*, is a significant necrotrophic disease of the leaves and glumes of wheat (*Triticum aestivum* L.). Disease epidemics, which when severe may cause yield losses up to 50%, are most common in wheat production regions of Australia, parts of northern Europe, North Asia, and the U.S. [1, 2]. In the U.S., SNB occurs alongside Septoria tritici blotch (caused by *Zymoseptoria tritici*) and/or tan spot (*Pyrenophora tritici-repentis*) in the western, moist areas of the Pacific Northwest; the upper Plains states of North Dakota, South Dakota, and Minnesota; and the states

**Data Availability Statement:** All relevant data are available on Figshare: https://doi.org/10.6084/m9.figshare.19372457.

**Funding:** This project was supported by the Agriculture and Food Research Initiative Competitive Grants 2017-67007-25939 and 2022-68013-36439 (WheatCAP) from the USDA National Institute of Food and Agriculture. The funders had no role in study design, data collection, and analysis, decision to publish, or preparation of the manuscript.

**Competing interests:** The authors have declared that no competing interests exist.

adjacent to the Great Lakes [3–5]. By contrast, to the east of the Appalachian Mountains, *P. nodorum* is both ubiquitous and greatly predominant over all other wheat leaf blight pathogens, making that environment ideal for field studies of SNB resistance and epidemiology [6].

There is a high degree of genetic diversity within and between *P. nodorum* populations from major wheat growing regions, in part due to approximately annual cycles of sexual reproduction [7]. Under warm moist conditions, the pathogen infects leaves, stems, and glumes directly through epidermal tissues. Lesions on lower leaves of plants are dark chocolate-brown initially and eventually develop into tan, lens- or irregular-shaped lesions with a yellow halo containing pycnidia that produce the asexual pycnidiospores. With the aid of heavy, splashy rainfall, glume infections may occur as pycnidia release conidia that splash up and infect grain heads, decreasing photosynthetic capacity of heads, peduncle, and flag leaf [8]. Infection occurs over a wide temperature range in wet and humid conditions, and stubble is often the primary source of inoculum for infection in subsequent years [9].

Genes controlling spike and foliar resistance to SNB may segregate independently [10–12]. Both seedling and adult plant resistance to SNB are quantitative and involve genes with minor additive effects, highly affected by environmental conditions [1, 11]. Considerable genotype × environment interaction is also expected in any genetic analysis for adult plant resistance to SNB, making efforts to identify resistance genes challenging. Several studies have identified leaf and glume resistance QTL accounting for less than 20% of the phenotypic variation [9]. For adult plant resistance to leaf blotch, QTL have been identified on wheat chromosomes 1A, 1B, 2A, 2D, 3A, 3B, 4B, 5A, 5B, 7B and 7A [12–19], and for glume blotch resistance, on 2A, 2B, 2D, 3A, 3B, 4A, 4B, 5A, 5B, 6B, 7A and 7D [12–14, 19–22].

Nine *Stagonospora nodorum necrosis* (*Snn*) genes that interact with matching necrotrophic effectors (NE) have been reported on wheat chromosome arms 1AS, 1BS, 2DS, 2DL, 4BL, 5BS, 5DS, 5BL, and 6AL [9, 23, 24]. These host genes were detected as quantitative trait loci (QTL) using bi-parental mapping populations and infiltration of *P. nodorum* culture filtrates in seedlings. If available pathogen populations carry the matching NE, an *Snn* gene may increase the severity of SNB in what is known as an "inverse gene-for-gene" interaction [24].

Breeding for SNB resistance in wheat is a challenging task due to complex quantitative genetic control and environmental influence. Variation in heading date and plant height are important determinants in breeding and selection; for example, wheat lines could be misclassified as SNB-resistant specifically if they are tall, late maturing genotypes that escape disease infection [9]. Some studies showed that genetic factors controlling heading date and plant height were actually linked to genes controlling SNB response rather than pleiotropic effects of agronomic characteristics affecting disease assessment [14, 16]. Markers identified for plant morphology and phenology in wheat include the semi-dwarf gene *Rht*, which controls variation for plant height, and vernalization gene *Vrn1* and photoperiod response gene *Ppd1*, that are major determinants of heading and flowering timing [25–30]. These diagnostic markers can add important value by allowing discrimination between linked resistance and pleiotropy caused by genes controlling plant height and heading date [9]. The objective of this study was to identify QTL associated with SNB leaf and glume resistance in a double haploid (DH) wheat population. We also investigated a QTL mapping approach that utilized DNA markers to account for differences in morphology and phenology.

## Materials and methods

### Plant material

A population of 124 DH lines was developed from the cross between the soft red winter wheat (SRWW) cultivar AGS2033, the moderately susceptible parent, and the winter breeding line

GA03185-12LE29, the moderately resistant parent. Both parents were developed by the University of Georgia (UGA). The population was designated the GADH population and subjected to screening for SNB resistance in the field at Raleigh, NC, along with the susceptible SRWW cultivar Jamestown as a control.

AGS2033 (96229/4/AGS2000*3/931433//PIO2684/*3AGS2000/3/AGS2000) was released in 2015 as a commercial cultivar and GA03185-12LE29 (97173-1-B//AGS2000*2/84202) is maintained by the UGA small grains breeding program [31]. AGS2033 is awned, has medium maturity, possesses two genes conferring photoperiod insensitivity (*Ppd-A1a.1* and *Ppd-D1a*), and requires long periods of vernalization to flower. AGS2033 carries the Robertsonian translocation of the short arm of rye chromosome 1R joined with the arm of wheat chromosome 1A (t1RS·1AL) and dwarfing allele *Rht-B1b*.

GA03185-12LE29 is awned and possesses medium maturity, the photoperiod sensitive alleles *Ppd-A1b* and *Ppd-D1b*, and the early flowering *vrn-B1* allele from AGS2000. GA03185-12LE29 carries the Robertsonian translocation of the short arm of rye chromosome 1R joined with the arm of wheat chromosome 1B (t1RS·1BL).

## Field phenotyping

The GADH population was phenotyped for SNB resistance in the field in four successive experiments: 2017–18, 2018–19, 2019–20, and 2020–21 (hereafter referred to as 2018, 2019, 2020, and 2021). Genotypes were tested each year at the Lake Wheeler Road Field Laboratory near Raleigh, North Carolina. Each genotype was planted in a single 1.3-meter row, with 3 g of seed per row except in a few cases where seed availability was lower. There were two replicate blocks each year, and genotypes were randomized within replicates. The field was conventionally plowed before planting, and plots were established with a headrow planter (Wintersteiger, Inc., Salt Lake City, UT). Planting occurred within the normal range of dates for the region, on 3 November 2017, 24 October 2018, 1 November 2019 and 20 October 2020. Standard practices were employed for management of fertility and weeds, but no fungicides were applied.

Inoculation was provided by dispersing naturally infected wheat straw at Zadoks growth stage 25–29 [32], prior to stem elongation but after plants were tillered sufficiently to avoid smothering by the straw [5]. The straw was sourced from commercial wheat fields in North Carolina, and therefore constituted a reservoir of a large, diverse set of isolates that represented the broader *P. nodorum* population of North Carolina. Each year, wheat straw was obtained from a single commercial farm. The straw was baled immediately following grain harvest and stored under cover until it was transported to the field for inoculation. Straw was applied to the plots at a rate of one rectangular bale per 40 plots (one headrow tray), with bale dimensions being approximately 1 meter long, 350–400 mm high, and 460 mm wide. The straw was spread evenly on plots by hand on 31 January 2018, 6 February 2019, 17 February 2020 and 8 February 2021.

During heading and anthesis, large-droplet irrigation was applied to enhance disease development and ensure inoculum dispersal to all upper plant parts. The total of rainfall and irrigation in the 40 days prior to disease assessment was 47 cm, 30 cm, 39 cm and 38 cm in the four years, respectively.

In 2018, nursery conditions were conducive to the development of a strong SNB epidemic; other diseases were minimal, and there was a small amount of freeze injury. In 2019, nursery conditions and disease development were also good, with no important confounding factors. In 2020, cool spring temperatures and low relative humidity resulted in a relatively light SNB epidemic. In 2021, low relative humidity again resulted in a relatively mild epidemic.

Preliminary QTL analysis of the 2018 and 2019 data identified SNB QTL coinciding with markers for flowering time genes. Days to heading (HD) was then assessed in the 2020 and 2021 experiments and was recorded as day of year when 50% of spikes in a row had fully emerged. Plant height (PH) was also recorded only in the 2020 and 2021 experiments, and height of plants was measured from the soil surface to the tip of spikes, excluding awns.

Disease severity was evaluated by the same assessor in all cases and was rated on a whole-plot basis at Zadoks growth stage 75–77 (medium to late milk stage of grain filling) [32]. Assessment dates were 19 May 2018, 10 May 2019, 22 May 2020, and 19 May 2021. Foliar and glume symptoms were rated separately. For foliar symptoms, a 1–9 scale was used, with 1 being the lowest level of disease observed and 9 the highest; for glume symptoms, the scale was 0–9.

## Marker development and high-resolution map construction

The DH population was subjected to genotyping by sequencing (GBS) for single nucleotide polymorphism (SNP) discovery according to Poland and Rife [33]. DNA was extracted from tissue collected from 10-day-old plants using the sbeadex plant maxi kit (LGC Genomics LLC, Teddington, UK). Ninety-six individual samples were barcoded, pooled into a single library, and sequenced on an Illumina HiSeq 2500. Sequencing reads were aligned to the International Wheat Genome Sequencing Consortium (IWGSC) RefSeqv2.0 assembly (https://wheat-urgi.versailles.inra.fr/Seq-Repository/Assemblies) using the Burrows-Wheeler Aligner v.0.7.12 [34]. The alignment information was processed by TASSEL-5GBSv2 pipeline version 5.2.35 [35] for SNP calling. The data were filtered to retain SNP with ≤20% missing data, ≥5% minor allele frequency, and ≤10% of heterozygous calls per marker.

Additionally, the DH population was genotyped with KASP assays for *Vrn-B1*, *Ppd-A1* and *Ppd-D1* [36–38]. To control for missing data due to the poor alignment of rye-derived reads with the wheat reference genome, the short arms of t1RS·1AL and t1RS·1BL translocation chromosomes were replaced with co-dominant KASP assays, IWA8035 and IWA6110, predictive of the presence of t1RS·1AL and t1RS·1BL, respectively, under the assumption of no recombination in those genomic regions (S1 Table). Alignment of reads to the recently published rye genome was not used for variant calling because of duplication of the 1R short arm in the population [39].

A genetic map was constructed with the GBS-SNP and KASP markers using the MSTmap algorithm in the R/ASMap and R/qtl packages [40–42]. Filtering removed low-quality markers with an excess of missing values (≥15%), segregation distortion (Chi-square test; alpha = 0.01), and co-located markers (duplicated marker information) before map construction. A total of 2,659 markers remaining after filtering were used to construct the linkage map. Recombination frequencies were estimated in centiMorgans (cM) using the Kosambi mapping function. For each linkage group, a recombination plot (Heatmap) was drawn using R and standard functions.

## Data analysis and QTL analysis

Analyses of variance (ANOVA) were performed using the lme4 and lmerTest packages for R [43, 44]. Adjusted means for individual and combined years of disease ratings of leaf severity (LS) and glume severity (GS), as well as heading date (HD) and plant height (PH), were estimated using the emmeans package for R [45]. The adjusted means were subjected to the Shapiro-Wilk normality test.

Quantitative trait loci (QTL) analysis was performed using the composite interval mapping (CIM) and standard interval mapping (SIM) approaches, with the significance LOD

(logarithm of the odds) threshold for an alpha = 0.05 determined using 5,000 permutations with the R/qtl package [46]. When major developmental genes (*Ppd-A1*, *Ppd-B1* and *Vrn-B1*) were found to be significant for HD, PH and SNB resistance using CIM, markers targeting these genes were used as covariates in the QTL analyses using SIM, and the estimation of each QTL effect on the phenotype was done using the Haley-Knott regression method (function "fitqtl") in the R/qtl package [42].

## Results

### Phenotypic analysis

Significant differences in HD, PH, and leaf and glume severity (LS and GS) ratings were observed for the DH lines, AGS2033, GA03185-12LE29 and the susceptible control Jamestown (*P* < 0.001). The parent GA03185-12LE29 was significantly later and taller than the parent AGS2033, with HD of 107.4 vs 100.0 days and PH of 107.8 vs. 88.4 cm, respectively (*P* < 0.001; Fig 1). Across years, the parent GA03185-12LE29 displayed mean ratings of 3.8 and 0.4 for LS and GS, respectively, qualifying it as moderately resistant to both foliar and glume disease. The mean ratings of parent AGS2033 were 5.1 and 3.3 for LS and GS, respectively, making it moderately susceptible in both categories. In comparison, ratings of 6.4 for LS and 4.5 for GS were observed for susceptible cultivar Jamestown. The differences between parents in mean ratings for both leaf and glume severity were also highly significant (*P* < 0.001; Fig 1). The GADH population showed continuous variation for HD, PH, LS and GS.

Combined across years, the HD and PH dataset deviated significantly from a normal distribution (*P* < 0.01), suggesting that major QTL might be involved. The multi-year LS and GS

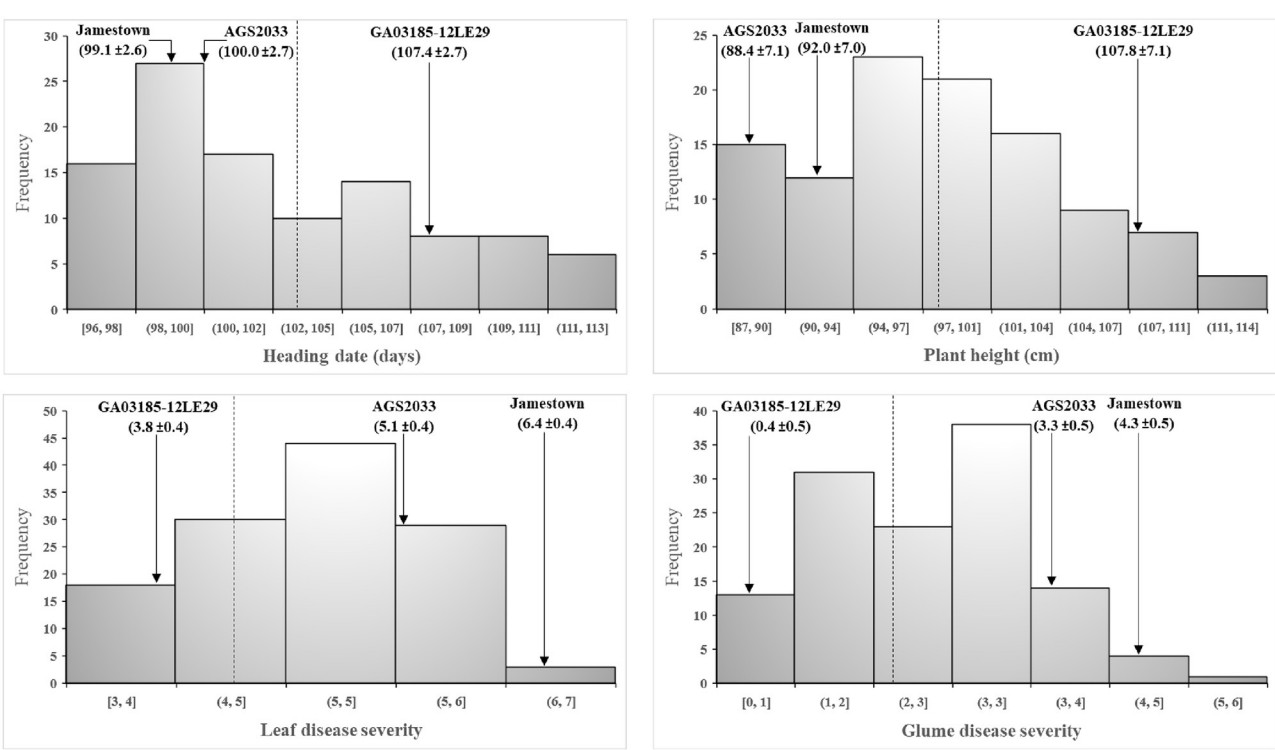

**Fig 1. Means across years for parents and DH lines of PH, HD, LS and GS.** Dashed vertical line indicates the GADH population mean for each trait. GA03185-12LE29 = moderate resistant parent. AGS2033 = moderate susceptible parent. Jamestown = susceptible control. Disease severity scale = 0–9.

**Table 1. Pearson's correlation coefficients (*r*) and mean estimates with standard errors for heading date, plant height, and Septoria nodorum blotch leaf and glume disease severity in the GADH population.**

| Trait | HD | PH | LS | Mean ± S.E. |
|-------|-----|-----|-----|-------------|
| **HD** | - | | | 102.7 ± 0.34 |
| **PH** | 0.69** | | | 98.0 ± 0.69 |
| **LS** | -0.80** | -0.69** | | 4.8 ± 0.09 |
| **GS** | -0.77** | -0.62** | 0.81** | 2.2 ± 0.10 |

Abbreviations: HD = heading date (days); PH = plant height (cm); LS = leaf severity (1–9 scale); GS = glume severity (0–9 scale). S.E. = standard error.

** *P*-value is less than 0.01.

data were normally distributed, and they were significantly correlated ($P < 0.01$; Table 1 and S1 Fig). There was a strong negative correlation ($P < 0.01$) between LS and both HD and PH ($r \leq -0.69$), and between GS and both HD and PH ($r \leq -0.62$), in the combined-year analysis (Table 1). Similar results were obtained for the within-year correlation analysis (S2 Table). These findings suggest that in the GADH population, late tall plants would be more resistant than early short plants (S2 Fig).

## Genetic map construction

A genetic map with 2,659 SNP markers was constructed for the GADH population. The map comprised 26 linkage groups assigned to 21 wheat chromosomes, of which markers aligning to 2B, 4A, 5A, 5D, 6D, and 7A comprised more than one linkage group and a linkage group for chromosome 3D was not assigned (S3 Table). The B genome had the largest number of markers, 1,400 (52.7%), followed by A with 1,025 (38.5%) and D with 234 (8.8%). Gaps greater than 30 cM were observed for chromosomes 2B (38.0 cM), 3B (35.1 cM), and 5A (36.0 cM) (S3 Table and S3 Fig). The linkage map spanned 2,936.3 cM, with 1,123.5, 1,435.8, and 410.2 cM in the A, B, and D genomes, respectively.

## QTL detection for HD, PH, and SNB disease resistance

Major QTL for HD, PH, and SNB resistance (LS and GS) were identified in the GADH population using the CIM approach (Table 2 and Fig 2). Heading date and plant height were evaluated in the Raleigh 2020 and 2021 environments. Three significant QTL associated with HD located in linkage groups 2A (*Ppd-A1*), 2D (*Ppd-D1*), and 5B (*vnr-B1*) explained 23.5, 44.2, and 29.1 percent of the phenotypic variation, respectively. Two significant QTL associated with PH located in linkage groups 2A and 2D explaining 16.7 and 45.7 percent of the phenotypic variation, respectively, co-located with HD QTL *Ppd-A1* and *Ppd-D1*. The *Ppd*-D1 locus accounted for almost half of the phenotypic variation for both traits (Table 2). For most of the HD and PH QTL, the allele from the moderate susceptible AGS2033 parent had a positive effect on PH and HD, except *Qncb.hd-5B* which had a negative allele effect on HD. This is consistent with the presence of photoperiod-insensitive alleles in AGS2033 at the *Ppd-A1* and *Ppd-D1* loci located on chromosomes 2A and 2D, respectively. The negative allele effect on the *Qncb.hd-5B* QTL was consistent with the presence in the GA03185-12LE29 parent of the *vrn-B1* allele, which is associated with earlier flowering after shorter periods of vernalization [38].

The CIM analysis across years detected three significant QTL associated with LS located in linkage groups 1A, 2A, and 2D that explained 7.3, 27.4, and 37.2 percent of the phenotypic variation, respectively (Table 2). Four significant QTL associated with GS located in linkage

**Table 2. Quantitative trait loci (QTL) in the GADH population associated with heading date, plant height, and SNB severity combined across years, using the composite interval mapping approach.**

| Trait | QTL | LG | cM | Confidence interval (cM) | LOD | *P*-value | PVE (%) | Allele effect[a] |
|-------|-----|----|----|--------------------------|-----|-----------|---------|-------------------|
| HD | *Ppd-A1* | 2A | 13 | 10–16 | 9.1 | 0.0000 | 23.5 | 2.1 |
| | *Ppd-D1* | 2D | 86 | 83–89 | 14.3 | 0.0000 | 44.2 | 2.9 |
| | *Qncb.hd-5B* | 5B | 153 | 152–157 | 11.4 | 0.0000 | 29.1 | -2.3 |
| PH | *Ppd-A1* | 2A | 13 | 10–16 | 4.8 | 0.0034 | 16.7 | 2.7 |
| | *Ppd-D1* | 2D | 86 | 83–89 | 6.0 | 0.0004 | 45.7 | 4.4 |
| LS | *Qncb.snl-1A* | 1A | 25 | 22–28 | 5.0 | 0.0006 | 7.3 | 0.3 |
| | *Ppd-A1* | 2A | 13 | 10–16 | 8.0 | 0.0000 | 27.4 | -0.5 |
| | *Ppd-D1* | 2D | 86 | 83–89 | 13.3 | 0.0000 | 37.2 | -0.6 |
| GS | *Qncb.sng-1A* | 1A | 0 | 0–3 | 4.7 | 0.0020 | 12.7 | 0.4 |
| | *Qncb.sng-2A* | 2A | 34 | 30–36 | 7.5 | 0.0000 | 32.1 | -0.7 |
| | *Ppd-D1* | 2D | 86 | 83–89 | 7.8 | 0.0000 | 31.9 | -0.7 |
| | *Qncb.sng-5B* | 5B | 153 | 143–182 | 4.5 | 0.0022 | 14.0 | 0.5 |

Abbreviations: LG = linkage group; cM = centimorgan; HD = heading date (days); PH = plant height (cm); LS = leaf severity (1–9 scale); GS = glume severity (0–9 scale). LOD = logarithm of the odds at 0.05 probability level, using a 5,000-iteration permutation test; PVE = percentage of phenotypic variance explained.
[a]Estimated allele effect reported in terms of the AGS2033 parent. The estimated allele effect unit corresponds to each trait unit.

groups 1A, 2A, 2D, and 5B explained 12.7, 32.1, 31.9, and 14.0 percent of the phenotypic variation, respectively. Phenological QTL coincided with disease severity QTL on chromosomes 2A, 2D, and 5B, as follows. On chromosome 2A, the major QTL *Ppd-A1* for HD and PH co-located with the SNB QTL for LS (Fig 2). For GS, by contrast, while *Ppd-A1* was near the 2A QTL, the photoperiod locus was not included in the confidence interval (CI) for the GS-linked marker *Qncb.sng-2A* (Fig 2). On chromosome 2D, the HD and PH QTL that were associated with *Ppd-D1* at position 86 cM also co-located with the SNB QTL for both LS and GS.

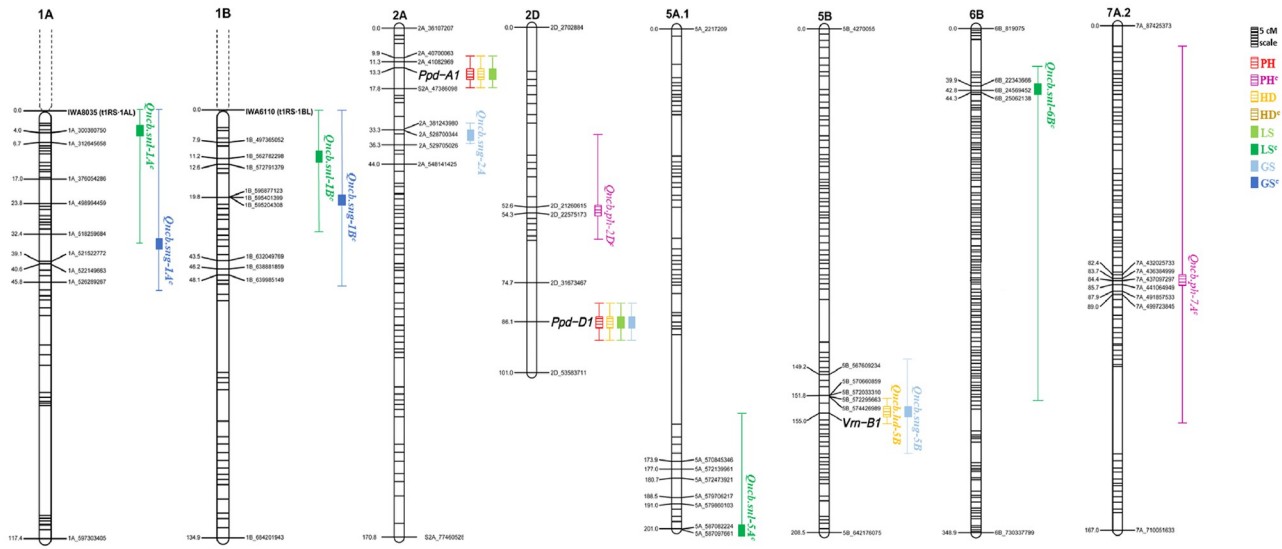

**Fig 2. Genetic map location of QTL detected in this study.** QTL locations and interval sizes are indicated by bars and brackets, respectively, on the right-hand side of each chromosome, and are based on the genetic marker information in Tables 2 and 3. [c] = covariate in standard interval mapping analysis; numbers after the underscore correspond to physical positions based on Chinese Spring RefSeq v2.0.

Similarly, on chromosome 5B, a major HD QTL *Qncb.hd-5B* associated with the *Vrn-B1* locus co-located with the 5B QTL for GS. GA03185-12LE29 alleles located in linkage groups 2A and 2D that had a positive allele effect on HD and PH had a negative allele effect on both LS and GS. Conversely, the GA03185-12LE29 allele at *Qncb.hd-5B* was associated with early heading with a positive effect on GS. This suggests that these QTL influenced SNB resistance through their effect on heading date and also plant height. Overall, the QTL located on linkage groups 2A, 2D, and 5B combined accounted for the majority of the variation for HD (96.8%), PH (62.4%), LS (64.6%), and GS (78.0%) in the GADH population (Table 2). The QTL on 1A was of smaller effect and was not associated with phenology.

In the within-year QTL analysis using CIM, *Ppd-A1* explained approximately 20% of the phenotypic variation for HD20, HD21, and PH20, and had no significant effect on PH21 (S4 Table). *Ppd-D1* explained 34.0 and 42.5 percent of the phenotypic variation for HD20 and HD21, respectively, and it was in close proximity to *Qncb.ph-2D*, at position 88 cM, for PH20 and PH21. *Qncb.hd-5A* and *Qncb.ph-5A* were located in close proximity with overlapping CIs and explained 7.9% and 4.0% of the phenotypic variation for HD20 and PH20, respectively. *Qncb.hd-5B* explained 12.2% and 36.1% of the phenotypic variation for HD20 and HD21, respectively.

The within-year CIM analysis also detected two major QTL for SNB resistance in linkage groups 2A and 2D, associated with the *Ppd-A1* and *Ppd-D1* loci, respectively (S5 Table). In 2018 and 2019, a QTL associated with *Ppd-A1* explained a large percentage (20.1% to 27.8%) of the phenotypic variation for LS, and in 2019 the same QTL explained 29.6% of GS variation. In addition, in 2021 a QTL on 2A for LS (*Qncb.snl-2A*) explaining 18.3% of the phenotypic variation was detected at 25cM. A QTL for GS (*Qncb.sng-2A*) explaining 18.2% to 27.7% of the phenotypic variation was also detected proximal to *Ppd-A1*, with peaks at 26 cM and 31 cM for GS18 and GS21, respectively. The 2D QTL explained a larger percentage (19.4%-41.4%) of the phenotypic variation for LS: for LS18 and LS21, the QTL co-located with *Ppd-D1* at position 86 cM, while for LS19 and LS20, it was detected at 88 cM. In addition, near the *Ppd-D1* locus at 85–88 cM, the QTL *Qncb.sng-2D* explained 35.7% to 40.9% of the phenotypic variation in GS in 2018 and 2019.

In linkage group 5B, the within-year analysis detected QTL (*Qncb.snl-5B* and *Qncb.sng-5B*) associated with SNB resistance for LS19, LS21, GS18 and GS19, and located near the *Vrn-B1* locus (S5 Table). The *Qncb.snl-5B* explained 8.9% to 22.9% of the phenotypic variation for leaf SNB resistance, and the *Qncb.sng-5B* explained 13.6% to 14.7% of the phenotypic variation for glume SNB resistance. Further, two QTL in linkage group 1A (*Qncb.snl-1A* and *Qncb.sng-1A*) were associated with SNB resistance for LS18, LS20, GS18 and GS20. The *Qncb.snl-1A* explained 12.0% to 12.2% of the phenotypic variation for foliar SNB resistance, and *Qncb.sng-1A* explained 17.8% to 20.4% of the phenotypic variation for glume resistance. Additionally, QTL were detected for LS only in 2019 in linkage groups 5A.1 (*Qncb.snl-5A*) and 6B (*Qncb.snl-6B*) at positions 196 cM and 49 cM, respectively. These QTL explained up to 14% of the phenotypic variation for LS19.

## QTL analysis using flowering time genes as covariates

Given the large and highly significant correlation of HD with LS and GS in the population and detection of large-effect QTL for both phenology and disease reaction associated with the *Ppd-A1*, *Ppd-D1* and *Vrn-B1* loci, a standard interval mapping (SIM) approach was investigated using the marker data for these three major flowering time genes as covariates. QTL detected in these analyses were designated with a superscript "c" (Table 3, Fig 2, S4 and S5 Tables). Once the effects of the major HD genes *Ppd-D1*, *Ppd-A1* and *Vrn-B1* were accounted for, no

**Table 3. Quantitative trait loci (QTL) in the GADH population associated with heading date, plant height, and disease severity combined across years, using the standard interval mapping (SIM) approach with three major phenology genes as covariates for SNB resistance.**

| Trait | QTL | LG | cM | Confidence interval (cM) | LOD | *P*-value | PVE (%) | Allele effect[a] |
|---|---|---|---|---|---|---|---|---|
| **PH** | *Qncb.ph-2D*[c] | 2D | 53 | 32–63 | 4.6 | 0.0044 | 33.2 | 3.6 |
| | *Qncb.ph-7A*[c] | 7A.2 | 84 | 8–130 | 3.3 | 0.0390 | 3.8 | 1.3 |
| **LS** | *Qncb.snl-1A*[c] | 1A | 4 | 0–36 | 5.7 | 0.0000 | 10.2 | 0.3 |
| | *Qncb.snl-1B*[c] | 1B | 11 | 0–30 | 3.3 | 0.0322 | 6.7 | 0.3 |
| | *Qncb.snl-5A*[c] | 5A.1 | 201 | 10–201 | 3.5 | 0.0216 | 4.0 | -0.2 |
| | *Qncb.snl-6B*[c] | 6B | 43 | 35–279 | 3.7 | 0.0146 | 11.8 | 0.3 |
| **GS** | *Qncb.sng-1A*[c] | 1A | 36 | 0–48 | 7.2 | 0.0000 | 11.8 | 0.4 |
| | *Qncb.sng-1B*[c] | 1B | 20 | 0–51 | 4.6 | 0.0022 | 5.6 | 0.3 |

Abbreviations: LG = linkage group; cM = centimorgan; HD = heading date (days); PH = plant height (cm); LS = leaf severity (1–9 scale); GS = glume severity (0–9 scale). LOD = logarithm of the odds at 0.05 probability level, using a 5,000-iteration permutation test; PVE = percentage of phenotypic variance explained.

[a]Estimated allele effect reported in terms of the AGS2033 allele. The estimated allele effect unit corresponds to each trait unit.

[c] = covariate.

further QTL for heading date were detected. However, a significant QTL associated with PH located in linkage group 2D, *Qncb.ph-2D*[c], was revealed at 53 cM, accounting for 33.2% of the variation in PH (Table 3). This locus was distal to the PH QTL associated with *Ppd-D1* locus (86 cM), suggesting segregation of a separate PH QTL on 2D in the population (Fig 2). A significant QTL of small effect on linkage group 7A.2 (84 cM) was also detected explaining 3.8 percent of the phenotypic variation for PH in the analysis including covariates (Table 3).

This new analysis using combined-year means for LS and GS detected significant QTL on 1A and 1B, along with QTL in linkage groups 5A.1 and 6B that were detected only in single environment analyses using CIM (Table 3, Fig 2 and S5 Table). The QTL *Qncb.snl-1A*[c], *Qncb.snl-1B*[c], *Qncb.snl-5A*[c] and *Qncb.snl-6B*[c] explained 10.2, 6.7, 4.0 and 11.8 percent of the phenotypic variation in LS, respectively. Significant QTL associated with GS located in linkage groups 1A (*Qncb.sng-1A*[c], 17 cM) and 1B (*Qncb.sng-1B*[c], 20 cM) explained 11.8 and 5.6 percent of the phenotypic variation, respectively. The QTL had a positive allele effect on LS and GS, suggesting alleles from the resistant parent GA03185-12LE29 were associated with increased disease resistance, except the QTL located in linkage group 5A.1 that had a negative allele effect for LS (Table 3).

The within-year QTL analysis using SIM with covariates detected *Qncb.hd-5A*[c] for HD20 and a new locus *Qncb.hd-5B*[c] for HD21 in linkage groups 5A.1 and 5B, respectively (S4 Table). For PH20, the analysis also detected *Qncb.ph-2D*[c] at position 40 cM and *Qncb.ph-5A*[c] at position 181 cM in linkage groups 2D and 5A.1, respectively. The QTL in linkage group 1A (*Qncb.snl-1A*[c] and *Qncb.sng-1A*[c]) associated with SNB resistance were detected for LS18, LS20, GS18 and GS20 (S5 Table). Two QTL were revealed in linkage group 1B (*Qncb.snl-1B*[c] and *Qncb.sng-1B*[c]) for LS20 and GS20, explaining up to 12% of the phenotypic variation for both traits. The *Qncb.snl-1B*[c] was distal to the *Qncb.sng-1B*[c] at positions 11 cM and 46 cM, respectively. One QTL at linkage group 5A.1 (*Qncb.snl-5A*[c] at position 201 cM), and one QTL at linkage group 5B (*Qncb.sng-5B*[c] at position 154 cM) were detected for LS19 and GS18, respectively. A QTL was detected in linkage group 6B (*Qncb.snl-6B*[c]) for LS19 and LS20, explaining up to 14% of the phenotypic variation in both years.

Regardless of the analytical approach, the QTL *Qncb.snl-1A*[c] (CI 0–36 cM) was consistently associated with LS within and across years, and the QTL *Qncb.sng-1A*[c] (CI 0–48 cM) was consistently associated with GS within and across years (Table 3 and Fig 2). On the other hand,

*Qncb.snl-1B^c* and *Qncb.sng-1B^c*, associated with LS and GS respectively on linkage group 1B, were revealed only when SIM with covariates was performed. Additionally, *Qncb.snl-6B^c*, with a large CI of 36–279 cM, was consistently associated with LS within and across years when HD covariates were used in the SIM analyses (Table 3, Fig 2 and S5 Table).

## Discussion

SNB resistance is an important breeding target in wheat cultivar development in the spring and winter regions of the U.S, as well as other parts of the world. Avoiding host sensitivity (*Snn*) genes that match necrotrophic effectors (NE) produced by local *P. nodorum* populations is a goal of many wheat breeders [9, 24]. However, the actual role of NE relative to other types of *P. nodorum* genes in determining wheat resistance levels in the field is not well understood [6]. Numerous studies found that field resistance to SNB was controlled by several QTL with small, additive effects that are not known to correspond to characterized *Snn* genes [12, 47]. In this study, we focused on investigating the field reaction of the GADH population AGS2033 × GA03185-12LE29 at the adult plant stage to detect genomic regions associated with SNB resistance in winter wheat. Plants of breeding line GA03185-12LE29 exhibited very little SNB glume blotch in the field and were moderately resistant to leaf symptoms. Winter wheat cultivar AGS2033 had more intermediate reactions to both leaf and glume blotch. However, these lines also differed in final PH and days to heading (Fig 1). Therefore, our analyses also explored the effect on SNB reaction of major genes for plant development segregating in the population.

### Co-location of major effect HD and PH variants with SNB resistance QTL

Our marker analysis indicated that major genes for HD were segregating in the population. In diagnostic KASP assays, AGS2033 and GA03185-12LE29 differed for alleles of the photoperiod loci *Ppd-A1* and *Ppd-D1* as well as winter alleles of the *Vrn-B1* locus. Both cultivars possessed the *Rht-D1b* semi-dwarfing allele. Initial analysis of phenotypic collected in 2018 and 2098 located SNB resistance QTL near major heading data variants, days to heading and PH were recorded for the GADH population in the 2020 and 2021 environments. Over years, *Ppd-A1*, *Ppd-D1* and a QTL in close proximity to *Vrn-B1* jointly explained 69% of the phenotypic variation for heading date. In 2020, a QTL explaining an additional 7.9% of variation in HD was also detected on the long arm of linkage group 5A.1 in the interval between 579.7 Mbp and 579.8 Mbp containing the *Vrn-A1* locus. These data are consistent with other studies reporting the large role of *PPD1* and *VRN1* genes on winter wheat flowering time in environments in the Mid-Atlantic and Southern U.S. [38, 48, 49].

Moreover, these heading date variants were associated with QTL for plant height in the GADH population. Plant height QTL were identified in close proximity to the photoperiod loci on 2A and 2D in both years and near *Qncb.hd-5A.1* in 2020; combined, they explained more than 60% of genetic variation for PH in the population, suggesting pleiotropic effects of the heading date loci on mature PH. Subsequent analyses using the *Ppd-A1*, *Ppd-D1* and *Vrn-B1* markers as covariates revealed the presence of an additional 2D QTL distal to *Ppd-D1* associated with PH that explained 33% of variation after accounting for the effect of these major genes. There are reports of PH loci on the short arm of chromosome 2D, most famously the *Rht8* locus (see below) that is linked to the *Ppd-D1* photoperiod insensitivity locus [50, 51].

Similar to HD and PH, our analysis identified QTL associated with variants of *Ppd-A1*, *Ppd-D1* and *Vrn-B1* for LS and GS. Plant height and HD were highly negatively correlated with LS and GS in the 2020 and 2021 environments, the years in which height and heading were assessed, and also with disease ratings in the previous years 2018 and 2019. Similar results

were reported by four other groups [14, 16, 52, 53]. They found significant, moderately negative correlation of heading date and plant height with leaf and glume disease severity within and across environments. These results suggest a significant role of heading date and plant height in the accurate assessment of SNB resistance in wheat.

In the context of the CIM approach, co-location of HD and PH QTL with SNB resistance QTL suggests that in the GADH population, loci underlying plant development are having a pleiotropic effect on disease severity or are tightly linked with SNB resistance loci (Fig 2). The *Ppd-D1* loci precisely co-located with both LS and GS QTL in chromosome 2D. Similarly, the *vrn-B1* allele, associated with earlier flowering after shorter periods of vernalization, was in close proximity to *Qncb.sng-5B* for GS. Other researchers also observed QTL for leaf and glume SNB resistance in chromosomes 2D and 5B in close proximity to the *Ppd-D1* and *Vrn-B1* loci, respectively. In chromosome 2D, several studies detected LS resistance QTL distal to the *Ppd-D1* loci [15, 54, 55] while other studies [12, 14] reported similar QTL proximal to the *Ppd-D1* loci. In addition, three studies reported GS QTL proximal to the *Ppd-D1* loci [12, 19, 21]. Most of these QTL explained a small portion of the resistance to SNB [56]. In those studies, LS and GS QTL did not co-locate with the *Ppd-D1* loci as we report in this study, although they were found in similar genomic regions. Once the flowering loci were taken into account, our QTL analysis did not locate additional QTL for SNB resistance on 2DS but did locate a plant height locus on the chromosome arm. This suggest that disease resistance associated with plant height was due primarily to pleiotropic effects of HD on both PH and disease. In our nursery, irrigation was applied specifically to reduce escapes due to tallness by regularly splashing inoculum up the canopy.

Although the *Ppd-A1* locus co-located with LS QTL on 2A, it was distal to the glume blotch resistance QTL *Qncb.sng-2A* in this study. Whether this reflects an additional locus on 2A for resistance to GS is not clear. Francki et al. (2018) used a high-density genetic map to discriminate between previously mapped SNB resistance QTL in hexaploid wheat [12]. They reported two QTL associated with LS in close proximity to the *Ppd-A1* locus. On the other hand, Shankar et al. (2008) and Jighly et al. (2016) reported GS QTL proximal to the *Ppd-A1* locus, perhaps similar to the one we identified in this study in chromosome 2A [14, 19].

When the major QTL controlling plant growth and developmental traits were used as covariates in the SIM analysis, none of the co-located QTL for SNB resistance on 2A, 2D and 5B were detected. This suggests that the flowering time loci are either having a pleiotropic effect on disease resistance or are closely linked to loci affecting disease such that their effect was not detectable in this analysis. Although plant growth and developmental traits do not trigger defense reactions, they provide mechanical barriers or escape mechanisms under SNB field outbreaks [13, 57]. In our study, around 28% of the DH lines were tall and late, and showed better resistance than short and early lines during SNB epidemics in Raleigh (S2 Fig). Our findings support the hypothesis that plant growth and development traits indeed provide escape mechanisms under biotic stresses in wheat.

Although major developmental loci segregating in the population were affecting disease levels, the population displayed normal continuous distribution for LS and GS reflecting the quantitative nature of the SNB resistance in wheat. Moreover, our analysis detected additional QTL associated with SNB resistance, indicating polygenic inheritance, which also was reported in previous studies on field SNB resistance in wheat [12, 17, 18].

## Identification of SNB resistance QTL of small effect

The use of major genes as covariates in a SIM approach was justified as *Ppd-A1*, *Ppd-D1* and *Vrn-B1* accounted for at least two-thirds of the variation in HD and PH in 2020 and 2021, and

likely played a large role in these traits in Raleigh during 2018 and 2019. Additionally, QTL near or co-located with the phenology genes on chromosomes 2A, 2D, and 5B accounted for 64.6% of variation for LS and 78% for GS in the combined analysis (Table 2). Use of the *Ppd-A1*, *Ppd-D1* and *Vrn-B1* markers as covariates revealed QTL for LS and GS on linkage groups 1A, 1B, 5A and 6B (Table 3). Of these, *Qncb.snl-1B^c*, *Qncb.snl-5A^c*, *Qncb.snl-6B^c* and *Qncb.sng-1B^c*, which were previously identified only in single environments, proved significant in multiple years and in the combined analysis. This suggest that QTL of smaller effect segregating in the population could be revealed by accounting for heading time with predictive markers for the underlying genes in environments where HD was not recorded.

Interestingly, the direction of effects for these smaller-effect SNB QTL indicated that alleles for reduced LS and GS were contributed by the more susceptible AGS2033 parent. The QTL *Qncb.snl-5A^c*, associated with reduced LS, co-located with *Qncb.hd-5A^c*, which was significant for HD only in the 2020 environment, with AGS2033 having the later heading allele (Table 3 and S4 Table). It is possible that this SNB QTL may also be related to heading time, as is it coincides with the *Vrn-A1* locus and we did not account for variation at *Vrn-A1* in our model. Markers within the *Vrn-A1* coding sequence were not polymorphic between parents of the population. However, effect of *Vrn-A1* on flowering in winter wheat can be associated with copy number variation in addition to sequence variation [58].

Each parent of the GADH mapping population possessed a translocation involving the short arm of rye chromosome 1R. The t1RS·1AL was detected in AGS2033 and t1RS·1BL in GA03185-12LE29. Under the assumption of no recombination between 1RS and wheat chromosome arms 1AS and 1BS, we used a single co-dominant KASP assay to represent the rye short arms in construction of our linkage maps. Confidence intervals for the QTL for LS and GS identified on chromosomes 1A and 1B included these KASP assays, indicating that *Qncb.snl-1A^c* and *Qncb.sng-1A^c* QTL are linked to the t1RS·1AL wheat-rye chromosomal translocation, and the *Qncb.snl-1B^c* and *Qncb.sng-1B^c* QTL are linked to the t1RS·1BL rye-wheat chromosomal translocation. Small amounts of recombination were noted between the KASP assays for 1RS and the peak QTL markers. However, the peak QTL markers and confidence intervals extended into distal portions of both chromosome arms 1AL and 1BL based on physical distances (Fig 2), reflecting suppressed recombination in proximal portions of the chromosome arms. This impedes our ability to resolve the QTL positions in this doubled haploid population. Other studies have reported QTL for field resistance to SNB on group one chromosomes [9, 12, 47], as well as the presence on 1A and 1B of *Snn* genes that interact with matching necrotrophic effectors [59–63]. However, these loci were located distally on 1AS and 1BS, whereas our peak QTL markers were located in the long arms of 1A and 1B. Given the broad QTL intervals observed, we are not able to speculate about the role of the wheat-rye translocations and their relationship with previously identified QTL in disease resistance/susceptibility in this population.

A new QTL associated with LS on linkage group 6B, *Qncb.snl-6B^c* at position 43 cM, was revealed after using major developmental loci segregating in our mapping population as covariates. This new QTL explained ~12% of the phenotypic variation for LS across years. Based on a recent review by Downie et al. 2021, no QTL associated with SNB adult plant resistance has been reported on chromosome 6B.

## Identification of *Rht8*

When we used the *Ppd-D1* locus as a covariate in the SIM analysis, additional QTL *Qncb.ph-2D^c* and *Qncb.ph-7A^c*, which explained 33.2% and 3.8% of the phenotypic variation for PH respectively, were revealed at 53 cM in linkage group 2D and 84 cM in linkage group 7A.2.

The $Qncb.ph-2D^c$ is likely the gibberellic acid responsive semi-dwarfing gene *Rht8* that was previously mapped by Liu et at. (2015) at position 20 cM and Czembor et al. (2019) at position 29 cM [57, 64]. DeWitt et al. (2021) also detected a PH QTL on 2DS at an interval of 23.3 to 32.2 Mbp, postulated as *Rht8*, when using *Ppd-D1* as a cofactor in QTL analysis of a bi-parental winter wheat population [49]. Likewise, $Qncb.ph-2D^c$ was located at an interval of 16.5 to 24.0 Mbp in our study.

Similar to the other Green Revolution semi-dwarfing genes, *Rht8* has been widely used in wheat breeding programs and reduces PH by up to 10 cm [50, 51]. *Rht8* and the photoperiod-insensitive *Ppd-D1* QTL often occur together, but *Ppd-D1* itself has also been reported to have pleiotropic effects on PH [65–67]. In a study conducted by Chebotar et al. (2013), the positions of *Rht8* and *Ppd-D1* on the 2D chromosome of winter bread wheat were clarified using SSR markers [68]. They reported a ~25 cM distance between *Rht8* and *Ppd-D1*. Similarly, in our GBS-SNP based genetic map, we observed a distance of ~31 cM between *Rht8* ($Qncb.ph-2D^c$) and *Ppd-D1* (Fig 2).

## Conclusion

Resistance to SNB results in part from variation in plant development and morphology, which could provide escape mechanisms during natural SNB epidemics, and resistance QTL of small additive effects linked to genetic factors controlling development traits. Here, we show that after accounting for the effects of major flowering time genes segregating in the population, six QTL of small effect providing resistance to SNB were revealed. Interestingly, the majority of small-effect QTL for LS and GS resistance to SNB were provided by the shorter and earlier-heading parent. Although the majority of taller and later-flowering lines displayed resistance to SNB through escape mechanisms, there were a few short-stature and early-heading lines displaying moderate resistance to SNB across environments. Further research is warranted on the potential of this germplasm as a source of moderate resistance to SNB. In the meantime, plant breeders will rely on recurrent screening of wheat nurseries to identify and retain favorable alleles for plant development combined with SNB resistance QTL for their specific environment.

## Supporting information

**S1 Fig. Scatter plots of combined-year leaf vs. glume disease severity in the GADH population.** Dashed vertical and horizontal lines indicate the GADH population mean for each trait. (PDF)

**S2 Fig. Scatter plots of combined-year plant height and heading date vs. combined-year leaf and glume disease severity in the GADH population.** Dashed vertical and horizontal lines indicate the GADH population mean for each trait. (PDF)

**S3 Fig. Genotyping by sequencing single nucleotide polymorphism (GBS-SNP) marker distribution for the 26 linkage groups of the GADH population derived from the AGS2033 x GA03185-12LE29 cross.** The vertical axis reflects genetic distance (cM = centiMorgans). The horizontal axis represents the linkage groups assigned to all 21 wheat chromosomes. (PDF)

**S1 Table. KASP assays for major genes and Robertsonian translocation used in this study.** Abbreviation: Chr = chromosome. (PDF)

**S2 Table. Pearson's correlation coefficients of heading date, plant height, leaf and glume disease severity in the GADH population.** Abbreviations: PH20 = plant height 2020 (cm); PH21 = plant height 2021 (cm); PH = combined-year plant height (cm); HD20 = heading date 2020 (days); HD21 = heading date 2021 (days); HD = combined-year heading date (days); LS18 = leaf severity 2018; LS19 = leaf severity 2019; LS20 = leaf severity 2020; LS21 = leaf severity 2021; LS, combined-year leaf severity; GS18 = glume severity 2018; GS19 = glume severity 2019; GS20 = glume severity 2020; GS21 = glume severity 2021; GS = combined-year glume severity. ** P-value is less than 0.01.
(PDF)

**S3 Table. Summary statistics of the genetic linkage map of the GADH population.** Abbreviations: LG = linkage group; cM = centiMorgans; Chr = chromosome; Ave. = average; Max. = maximum; bp = base pairs.
(PDF)

**S4 Table. Within-year QTL associated with heading date and plant height in the GADH population using the CIM and SIM with major genes as covariates.** Abbreviations: CIM = composite interval mapping; SIM = standard interval mapping; QTL = quantitative trait loci; LG = linkage group; cM = centimorgan; HD = heading day (days); PH = plant height (cm); HD20 = heading date 2020; HD21 = heading date 2021; PH20 = plant height 2020; PH21 = plant height 2021; LOD = logarithm of the odds at 0.05 level of probability, obtained through a 5,000-iteration permutation test; PVE = percentage of phenotypic variance explained by the QTL. [a]Estimated allele effect reported in terms of the AGS2033 parent. The estimated allele effect unit corresponds to each trait unit. [c] = covariate.
(PDF)

**S5 Table. Within year quantitative trait loci (QTL) associated with leaf disease severity (LS) and glume disease severity (GS) in the GADH population using the CIM and SIM with major genes as covariates.** Abbreviations: CIM = composite interval mapping; SIM = standard interval mapping; QTL = quantitative trait loci; LG = linkage group; cM = centimorgan; LS = leaf severity (1–9 scale); GS = glume severity (0–9 scale); LS18 = leaf severity 2018; LS19 = leaf severity 2019; LS20 = leaf severity 2020; LS21 = leaf severity 2021; GS18 = glume severity 2018; GS19 = glume severity 2019; GS20 = glume severity 2020; GS21 = glume severity 2021; LOD = logarithm of the odds at 0.05 level of probability, obtained through a 5,000-iteration permutation test; PVE = percentage of phenotypic variance explained by the QTL. [a]Estimated allele effect reported in terms of the AGS2033 parent. The estimated allele effect unit corresponds to each trait unit. [c] = covariate.
(PDF)

## Acknowledgments

The authors thank Kim Howell, Jared Smith and Brian Ward for assistance with collection of genotypic data, and Logan Clark, Michael Elliott, Charlie Glover and the staff of the Lake Wheeler Road Field Laboratory for assistance with plot establishment, inoculation and irrigation.

## Author Contributions

**Conceptualization:** Gina Brown-Guedira, Christina Cowger.

**Data curation:** Luis A. Rivera-Burgos.

**Formal analysis:** Luis A. Rivera-Burgos.

**Funding acquisition:** Christina Cowger.

**Investigation:** Luis A. Rivera-Burgos.

**Resources:** Jerry Johnson, Mohamed Mergoum, Christina Cowger.

**Supervision:** Gina Brown-Guedira.

**Visualization:** Gina Brown-Guedira.

**Writing – original draft:** Luis A. Rivera-Burgos.

**Writing – review & editing:** Gina Brown-Guedira, Christina Cowger.

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
