## [Decision Letter · Decision Letter 0]

2 Feb 2022

PONE-D-21-34614Detection of small-effect QTL associated with the resistance to Septoria nodorum blotch in a hexaploid winter wheat populationPLOS ONE

Dear Dr. Rivera-Burgos,

Thank you for submitting your manuscript to PLOS ONE. After careful consideration, we feel that it has merit but does not fully meet PLOS ONE’s publication criteria as it currently stands. Therefore, we invite you to submit a revised version of the manuscript that addresses the points raised during the review process.

We look forward to receiving your revised manuscript.

Kind regards,

Aimin Zhang, Ph.D.

Academic Editor

PLOS ONE

Journal Requirements:

“This project was supported by the Agriculture and Food Research Initiative Competitive Grants 2017-67007-25939 (WheatCAP) from the USDA National Institute of Food and Agriculture.”

“This project was supported by the Agriculture and Food Research Initiative Competitive Grants 2017-67007-25939 (WheatCAP) from the USDA National Institute of Food and Agriculture.”

 “This project was supported by the Agriculture and Food Research Initiative Competitive Grants 2017-67007-25939 (WheatCAP) from the USDA National Institute of Food and Agriculture”

6. Please include a copy of tables 1 and 2 which you refer to in your text on page 10 and 11

Reviewers' comments:

Reviewer's Responses to Questions

**Comments to the Author**

1. Is the manuscript technically sound, and do the data support the conclusions?

Reviewer #1: No

2. Has the statistical analysis been performed appropriately and rigorously? 

Reviewer #1: Yes

3. Have the authors made all data underlying the findings in their manuscript fully available?

Reviewer #1: Yes

4. Is the manuscript presented in an intelligible fashion and written in standard English?

Reviewer #1: Yes

5. Review Comments to the Author

Reviewer #1: Dear Editor,

Rivera-Burgos et al report on the identification of QTLs associated with leaf and glume Septoria nodorum blotch (SNB) resistance using a DH population between AGS2033 (moderately susceptible) and GA03185-12LE29 (moderately resistant) based on the field data over four cropping seasons. The paper is well written and is of relevance as it studies a major disease affecting one of the crops that feed to world. I find particularly interesting how the authors dissect escape mechanisms due to phonological traits from “true” resistance QTL thought their analysis.

MAJOR COMMENTS

I have two main critiques to the submitted manuscript. From one side, there is only one day of scoring. Authors themselves argue in the discussion that heading date and plant height can play a critical role in the accurate assessment of SNB resistance in wheat. Considering that the parents to develop the DH population substantially differ for these two traits, I was wondering what is the bias (or not) imposed by a single day of scoring. I would agree with a single day of scoring in case the parents would be more similar for HD and PH. However, with those differences, I have my doubts. With a single day of scoring, it is impossible to get all the progeny evaluated at the same developmental stage, isn’t it?

Authors make use of CIM for their analysis. I am wondering if ICIM would give similar results. Moreover, HD and PH were only measured in two seasons. How the lack of HD and PH data for year 2018-19 affect the conclusion of this study?

From the other side, while results are nicely executed and explained, I am missing for the discussion a deeper reasoning compared to other studies to clearly state that the QTLs here found are actually novel. It looks like that author cannot clearly state that the 5 QTL with small effect are different QTLs or not from other QTLs reported in previous studies. This point need clear clarification or more convincing data/reasoning before eventual acceptance of the paper.

MINOR COMMENTS

Introduction

2nd paragraph. «Contributing to population structure” What do authors refer to?

I am missing few sentences about the relevance of doing this study on the GADH population. Is AGS2033 of special interest for the breeding community?

Material and Methods

Overall, M&M are very well written and explained. As said before, my major doubt is the timing of the disease severity measurement. Could author elaborate a little in this regard?

Results

1st paragraph, third line. Later? Does it refer to heading date? If so, please write it.

Section “Genetic Map Construction”

From were are coming the 2,659 SNP used to construct the linkage map? As these the polymorphic ones between the two parents of the DH population from the original 10,164 SNPs?

Section “QTL Detection for HD, PH, and SNB Disease Resistance”

2nd paragraph. Does this refer to the combined CIM analysis across years? It is not clear.

Discussion

Section “Co-location of Major Effect HD and PH Variants with SNB Resistance QTL”

5th paragraph. Authors need to reason is Qncb.sng-2A is a new LS QTL or not compared to Francki et al., 2018, Shankar et al., 2008, Jighly et al., 2016.

Section “Identification of SNB Resistance QTL of Small Effect”

2nd paragraph. It is unclear is Qncb.snl-5A is an actual QTL for SNB resistance or it due to the presence of Vrn-A1.

Paragraph 4th. A deeper discussion on Qncb.snl-6Bc compared to Eriksen et al., 2003 is needed. Could pedigree help?

Section “Identification of Rht8”

Overall, authors don’t prove enough data/reasoning that Qncb.ph-7Ac is different from the one reported by Gao et al., 2015. As suggested before.

Figures

Figure 2. Do the numbers after the underscore correspond to physical positions? I guess it is based on CS reference genome. If so, please indicate in the legend. Besides, for the case of 1A and 1B, it wouldn’t make more sense to use the rye reference genome?

S1 Fig. include correlation factor

6. PLOS authors have the option to publish the peer review history of their article (what does this mean?). If published, this will include your full peer review and any attached files.

Reviewer #1: No

---

## [Author Response · Author response to Decision Letter 0]

18 Mar 2022

Reviewers' comments:

Reviewer #1: Dear Editor,

Rivera-Burgos et al report on the identification of QTLs associated with leaf and glume Septoria nodorum blotch (SNB) resistance using a DH population between AGS2033 (moderately susceptible) and GA03185-12LE29 (moderately resistant) based on the field data over four cropping seasons. The paper is well written and is of relevance as it studies a major disease affecting one of the crops that feed to world. I find particularly interesting how the authors dissect escape mechanisms due to phonological traits from “true” resistance QTL thought their analysis.

MAJOR COMMENTS

I have two main critiques of the submitted manuscript.

1. From one side, there is only one day of scoring. Authors themselves argue in the discussion that heading date and plant height can play a critical role in the accurate assessment of SNB resistance in wheat. Considering that the parents to develop the DH population substantially differ for these two traits, I was wondering what is the bias (or not) imposed by a single day of scoring. I would agree with a single day of scoring in case the parents would be more similar for HD and PH. However, with those differences, I have my doubts. With a single day of scoring, it is impossible to get all the progeny evaluated at the same developmental stage, isn’t it?

Response:

HD: It’s true that on the day of scoring, lines were at slightly different stages of maturity. However, the difference between the parents was 7 days which, while not insignificant, is not that big either. HD varied continuously in this population, so the approach that I think the reviewer is advocating would have been to create perhaps 3 maturity bins for the population, and rate members of a given bin on a single day. Had the parents’ difference in HD been greater – and we sometimes see differences of 10-14 days or more -- that would have been important. We would make two points:

 1. Maturity differences collapse as the season progresses after heading. The 7-day difference at heading had already shrunk to 3 or 4 days by the time of rating, which was not at heading but rather at late milk /early dough.

2. It wasn’t feasible logistically to rate each line a fixed number of days after HD, which varied continuously in this population. So binning by HD would have been required, which effectively might have meant rating on perhaps 3 days instead of 1. We do not think this would have made a significant difference in the rating for each line or identification of QTL co-located with HD genes. This is not a disease that progresses that much from one day to the next; at that point in the epidemic, it’s a matter of lesions expanding, rather than new crops of lesions appearing.

Although both parents are semi-dwarf lines having the Rht-D1b allele, there was plant height variation in the population; we don’t know what is an alternative way to rate based on PH differences.

We have tried to clarify the relationship between heading date and plant height effects by adding additional text to the discussion section (page 21 & lines 435-440):

 “Once the flowering loci were taken into account, our QTL analysis did not locate additional QTL for SNB resistance on 2DS but did locate a plant height locus on the chromosome arm. This suggests that disease resistance associated with plant height was due primarily to pleiotropic effects of HD on both PH and disease. In our nursery, irrigation was applied specifically to reduce escapes due to tallness by regularly splashing inoculum up the canopy.”

2. Authors make use of CIM for their analysis. I am wondering if ICIM would give similar results. Moreover, HD and PH were only measured in two seasons. How does the lack of HD and PH data for year 2018-19 affect the conclusion of this study?

Response:

It is possible that Inclusive Composite Interval Mapping maybe also have identified SNB QTL not related to phenology. However, our SIM analysis using diagnostic markers as cofactors provided a method to account for variation in the correlated trait of heading date by identifying the underlying genes. We think that this adds to the understanding of QTL effects.

QTL for SNB reaction co-located with known determinants of heading date were identified during preliminary analysis of the first two years of data. This led to an assessment of heading date and plant height in the disease nursery during 2020 and 2021. We added a sentence to M&M (page 7 & lines 144-145) to clarify this point. 

“Preliminary QTL analysis of the 2018 and 2019 data identified SNB QTL coinciding with markers for flowering time genes. Days to heading (HD) was then assessed in the 2020 and 2021 experiments…..”

The genotypic data for the underlying causal polymorphisms in the heading date genes explained the majority of the HD variation in the population in both years, as noted at various places in the results (ie. Table 2) and in the discussion on page 22 & lines 464-466: 

“The use of major genes as covariates in a SIM approach was justified as Ppd-A1, Ppd-D1 and Vrn-B1 accounted for at least two-thirds of the variation in HD and PH in 2020 and 2021, and likely played a large role in these traits in Raleigh during 2018 and 2019.”

3. From the other side, while results are nicely executed and explained, I am missing for the discussion a deeper reasoning compared to other studies to clearly state that the QTLs here found are actually novel. It looks like that author cannot clearly state that the 5 QTL with small effect are different QTLs or not from other QTLs reported in previous studies. This point need clear clarification or more convincing data/reasoning before eventual acceptance of the paper.

Response:

Variation in heading day and plant height are constantly generated in breeding lines. But, how do we control for these traits' pleiotropic effects? How do reliably breed for resistance to SNB? How do avoid only selecting disease escapes genotypes? 

We showed evidence that HD pleiotropic effects can cause a misinterpretation of results as the major effect QTL for SNB resistance in our population were actually coinciding with HD genes. We are suggesting the change in title to emphasize this as the primary result of our analysis, rather than identification of unique small effect loci. 

While overall, our study suggests a robust approach of using morphological diagnostic markers to distinguish escapes (S2 Fig.) from SNB true resistance, we cannot say conclusively that the minor effect QTL are unique. We have not suggested this in the discussion.

MINOR COMMENTS

Introduction:

1. 2nd paragraph. «Contributing to population structure” What do authors refer to?

Response:

The sentence was vague; it has been adjusted to be clearer. Now, it reads (page 3 & lines 55-57) 

“There is a high degree of genetic diversity within and between P. nodorum populations from major wheat growing regions, in part due to approximately annual cycles of sexual reproduction [7].”

2. I am missing a few sentences about the relevance of doing this study on the GADH population. Is AGS2033 of special interest to the breeding community?

Response:

The interest for the breeding community focus on the high level of foliar and glume resistance to SNB exhibited by the breeding line GA03185-12LE29 in the 2012-2013 Eastern Septoria Nursery. By contrast the commercial cultivar AGS2033 exhibited moderate susceptibility. The parents displayed contrasting phenotypes for SNB resistance, making them ideal to develop a SNB segregating population for mapping purposes. (https://www.ars.usda.gov/southeast-area/raleigh-nc/plant-science-research/docs/nursery-reports/main/). 

Material and Methods:

3. Overall, M&M are very well written and explained. As said before, my major doubt is the timing of the disease severity measurement. Could author elaborate a little in this regard?

Response:

Please, see the answer to the first question of major comments above.

Results:

4. 1st paragraph, third line. Later? Does it refer to heading date? If so, please write it.

Response:

Yes, “later” refers to heading day as noted in the text (page 10 & lines 198-200): “The parent GA03185-12LE29 was significantly later and taller than the parent AGS2033, with HD of 107.4 vs 100.0 days and PH of 107.8 vs. 88.4 cm, respectively (P < 0.001; Fig 1).”

5. Section “Genetic Map Construction”: From where are coming the 2,659 SNPs used to construct the linkage map? As these the polymorphic ones between the two parents of the DH population from the original 10,164 SNPs?

Response:

By default, all GSB-SNP are polymorphic. The 2,659 SNP were the remaining SNPs after filtering for low-quality and co-segregating markers. We have attempted to clarify this point on pages 9 & lines 177-178:

“A genetic map was constructed with the GBS-SNP and KASP markers using the MSTmap algorithm in the R/ASMap and R/qtl packages [39–41]. Filtering removed low-quality markers with an excess of missing values (≥15%), segregation distortion (Chi-square test; alpha = 0.01), and co-located markers (duplicated marker information) before map construction. A total of 2,659 markers remaining after filtering were used to construct the linkage map.”

6. Section “QTL Detection for HD, PH, and SNB Disease Resistance”: 2nd paragraph. Does this refer to the combined CIM analysis across years? It is not clear.

Response:

This was clarified in the first sentence of the paragraph on page 13 & line 269:

“The CIM analysis across years detected three significant QTL…..”

Discussion

7. Section “Co-location of Major Effect HD and PH Variants with SNB Resistance QTL” 5th paragraph. Authors need to reason is Qncb.sng-2A is a new LS QTL or not compared to Francki et al., 2018, Shankar et al., 2008, Jighly et al., 2016.

Response:

The presence of heading date pleiotropic effect segregating in the GA DH population prevented us to determine whether Qncb.sng-2A reflects an additional locus on 2A resistance to GS. This QTL was not detected in SIM with covariates (Table 3). However, the literature reported several SNB QTL on 2A (Francki et al., 2018, Shankar et al., 2008, Jighly et al., 2016). 

8. Section “Identification of SNB Resistance QTL of Small Effect”: 2nd paragraph. It is unclear is Qncb.snl-5A is an actual QTL for SNB resistance or it due to the presence of Vrn-A1.

Response:

We are unable to determine this. The lack of polymorphism within the Vrn-A1 coding sequence and the co-location with Qncb.hd-5Ac, which affected HD20, prevented us from ruling out whether Qncb.snl-5Ac is an actual QTL for SNB or may also be related to heading time. We have attempted to clarify this by modifying the text (pages 22-23 & lines 476-483):

“The QTL Qncb.snl-5Ac, associated with reduced LS, co-located with Qncb.hd-5Ac, which was significant for HD only in the 2020 environment, with AGS2033 having the later heading allele (Table 3 and S4 Table). It is possible that this SNB QTL may also be related to heading time, as is it coincides with the Vrn-A1 locus and we did not account for variation at Vrn-A1 in our model. Markers within the Vrn-A1 coding sequence were not polymorphic between parents of the population. However, effect of Vrn-A1 on flowering in winter wheat can be associated with copy number variation in addition to sequence variation [57].”

9. Paragraph 4th. A deeper discussion on Qncb.snl-6Bc compared to Eriksen et al., 2003 is needed. Could pedigree help?

Response:

Eriksen et al., 2003 reported QTL associated with septoria tritici blotch (STB) resistance in the 6B linkage group. We cited Eriksen et al., 2003 by mistake. To avoid confusion, we have attempted to clarify this by modifying the text (page 24 & lines 505-507):

“A new QTL associated with LS on linkage group 6B, Qncb.snl-6Bc at position 43 cM, was revealed after using major developmental loci segregating in our mapping population as covariates. This new QTL explained ~12% of the phenotypic variation for LS across years. Based on a recent review by Downie et al. 2021, no QTL associated with SNB adult plant resistance has been reported on chromosome 6B.”

10. Section “Identification of Rht8” Overall, authors don’t prove enough data/reasoning that Qncb.ph-7Ac is different from the one reported by Gao et al., 2015. As suggested before.

Response:

Page 25 & lines 529-534

We have removed the paragraph referring to the QTL reported by Gao et al, 2015. This section is meant to focus on the separation of the 2DS plant height QTL from the Ppd-D1a effect when Ppd-D1 marker is used as a covariate in the analysis.

Figures

11. Figure 2. Do the numbers after the underscore correspond to physical positions? I guess it is based on CS reference genome. If so, please indicate in the legend.

Response:

The legend was modified as noted below:

“Fig 2. Genetic map location of QTL detected in this study. QTL locations and interval sizes are indicated by bars and brackets, respectively, on the right-hand side of each chromosome, and are based on the genetic marker information in Tables 2 and 3. c = covariate in standard interval mapping analysis; numbers after the underscore correspond to physical positions based on Chinese Spring RefSeq v2.0”

12. Besides, for the case of 1A and 1B, it wouldn’t make more sense to use the rye reference genome?

Response:

The alignment of short reads to the rye reference genome is complicated by the fact that the population segregated for both translocations. Variants in common between the different 1RS arms and the reference would be recorded as a single locus, complicating map construction. We have robust predictive KASP assays that are quite reliable at determining which lines possess either the t1RS.1AL or t1RS.1BL chromosomes, particularly in a bi-parental population. This was demonstrated in our work as the KASP markers were in close linkage with GBS SNP in the respective wheat long arms (Figure 2).

The following sentence was added on page 8 & line 171-173 to clarify this point: 

“Alignment of reads to the recently published rye genome was not used for variant calling because of duplication of the 1R short arm in the population (Rabanus-Wallace et al., 2021).”

13. S1 Fig. include correlation factor = correlation coefficient

Response:

We added r = 0.81.

---

## [Decision Letter · Decision Letter 1]

14 Apr 2022

PONE-D-21-34614R1Accounting for heading date gene effects allows detection of small-effect QTL associated with resistance to Septoria nodorum blotch in wheatPLOS ONE

Dear Dr. Rivera-Burgos,

Thank you for submitting your manuscript to PLOS ONE. After careful consideration, we feel that it has merit but does not fully meet PLOS ONE’s publication criteria as it currently stands. Therefore, we invite you to submit a revised version of the manuscript that addresses the points raised during the review process.

We look forward to receiving your revised manuscript.

Kind regards,

Aimin Zhang, Ph.D.

Academic Editor

PLOS ONE

Journal Requirements:

Reviewers' comments:

Reviewer's Responses to Questions

**Comments to the Author**

1. If the authors have adequately addressed your comments raised in a previous round of review and you feel that this manuscript is now acceptable for publication, you may indicate that here to bypass the “Comments to the Author” section, enter your conflict of interest statement in the “Confidential to Editor” section, and submit your "Accept" recommendation.

Reviewer #1: All comments have been addressed

Reviewer #2: (No Response)

2. Is the manuscript technically sound, and do the data support the conclusions?

Reviewer #1: Yes

Reviewer #2: Partly

3. Has the statistical analysis been performed appropriately and rigorously? 

Reviewer #1: Yes

Reviewer #2: Yes

4. Have the authors made all data underlying the findings in their manuscript fully available?

Reviewer #1: Yes

Reviewer #2: Yes

5. Is the manuscript presented in an intelligible fashion and written in standard English?

Reviewer #1: Yes

Reviewer #2: Yes

6. Review Comments to the Author

Reviewer #1: (No Response)

Reviewer #2: This manuscript reports the identification of resistance QTL to leaf and glume Septoria nodorum blotch (SNB) using a double haploid (DH) population derived from the cross between the moderately susceptible cultivar AGS2033 and the resistant breeding line GA03185-12LE29. And the authors identified major heading date (HD) and plant height (PH) variants on chromosomes 2A and 2D, co-located with SNB loci. Meanwhile, five minor QTL associated with adult plant resistance to SNB leaf and glume blotch were detected on 1A, 1B, and 6B linkage groups. The paper is written in standard English and SNB is a major disease affecting wheat. However, there are several results the authors should be carefully made.

1. the rise of SNB is similar to the yellow rust, and the temperature and the humidity had significant effect to the disease. As a result, the growth period may be one of the main reasons of the disease. Based on this, the phenotype identified in the manuscript may not accurate. As we detect the yellow rust, it was detected in several time, e.g. seedling stage and adult plant stage. Is it accurate to identify at only one time?

2. The authors use flowering time genes as covariates to detect SNB QTL. Is this suitable? The relationship of flowering time and the SNB is direct? and flowering time is the component of SNB?

3. “Each genotype was planted in a single 1.3-meter row, with 3 g of seed per row except in a few cases where seed availability was lower”. “3 g of seed per 1.3-meter” that means there may -70 seeds per 1.3 meter? Is this density too high for plant growth?

7. PLOS authors have the option to publish the peer review history of their article (what does this mean?). If published, this will include your full peer review and any attached files.

Reviewer #1: **Yes: **Javier Sánchez-Martín

Reviewer #2: No

---

## [Author Response · Author response to Decision Letter 1]

2 May 2022

PONE-D-21-34614R1

Overall, our study presents a simple and robust approach not currently reported in the literature to map resistance quantitative trait loci (QTL) to leaf and glume blotch. This approach uses the power of predictive markers associated with developmental traits to control for pleiotropic effects. We revealed SNB QTL on 1A, 1B, and 6B linkage groups. Although we are limited in our ability to draw conclusions about the relationship of these markers with previously reported QTL, the small-effect QTL detected in this study did not overlap with QTL associated with morphological and developmental traits, and thus are sources of resistance to SNB.

Journal Requirements:

Response:

All references listed are complete and correct. We have not cited retracted articles. Thank you. 

Reviewers' comments:

Comments to the Author

1. If the authors have adequately addressed your comments raised in a previous round of review and you feel that this manuscript is now acceptable for publication, you may indicate that here to bypass the “Comments to the Author” section, enter your conflict of interest statement in the “Confidential to Editor” section, and submit your "Accept" recommendation.

Reviewer #1: All comments have been addressed.

Reviewer #2: (No Response).

2. Is the manuscript technically sound, and do the data support the conclusions?

Reviewer #1: Yes

Reviewer #2: Partly

3. Has the statistical analysis been performed appropriately and rigorously?

Reviewer #1: Yes

Reviewer #2: Yes

4. Have the authors made all data underlying the findings in their manuscript fully available?

Reviewer #1: Yes

Reviewer #2: Yes

5. Is the manuscript presented in an intelligible fashion and written in standard English?

Reviewer #1: Yes

Reviewer #2: Yes

6. Review Comments to the Author

Reviewer #1: (No Response)

Reviewer #2: This manuscript reports the identification of resistance QTL to leaf and glume Septoria nodorum blotch (SNB) using a double haploid (DH) population derived from the cross between the moderately susceptible cultivar AGS2033 and the resistant breeding line GA03185-12LE29. And the authors identified major heading date (HD) and plant height (PH) variants on chromosomes 2A and 2D, co-located with SNB loci. Meanwhile, five minor QTL associated with adult plant resistance to SNB leaf and glume blotch were detected on 1A, 1B, and 6B linkage groups. The paper is written in standard English and SNB is a major disease affecting wheat. However, there are several results the authors should be carefully made: 

1. The rise of SNB is similar to the yellow rust, and the temperature and the humidity had significant effect to the disease. As a result, the growth period may be one of the main reasons of the disease. Based on this, the phenotype identified in the manuscript may not accurate. As we detect the yellow rust, it was detected in several time, e.g. seedling stage and adult plant stage. Is it accurate to identify at only one time?

Response: 

We appreciate the reviewer’s concern. The latent period of SNB is usually longer than that of the YR epidemics with which the reviewer is familiar. Thus, the rate of YR disease development is typically much higher than the rate of SNB development. It is common to rate SNB epidemics as disease severity (DS, a single point in time) rather than area under disease progress curve (AUDPC, integrated over time). This reflects that SNB typically only causes economic loss if it heavily affects flag leaves and glumes, while early establishment of YR can cause severe economic loss through stunting and loss of photosynthate for grain fill. No changes were made in response to this comment.

2. The authors use flowering time genes as covariates to detect SNB QTL. Is this suitable? The relationship of flowering time and the SNB is direct? And flowering time is the component of SNB?

Response: 

It was suitable to investigate the relationship between QTL for SNB reaction and determinants of maturity; i.e., to determine whether SNB response was simply due to heading date. In fact, QTL for SNB reaction co-located with known determinants of heading date were identified during preliminary analysis of the first two years of data. This led to an assessment of heading date and plant height in the disease nursery during 2020 and 2021.

This was already explained in the several places:

Page 7 & lines 142-145: “Preliminary QTL analysis of the 2018 and 2019 data identified SNB QTL coinciding with markers for flowering time genes. Days to heading (HD) was then assessed in the 2020 and 2021 experiments…..”

Page 21 & lines 433-438: “Once the flowering loci were taken into account, our QTL analysis did not locate additional QTL for SNB resistance on 2DS but did locate a plant height locus on the chromosome arm. This suggests that disease resistance associated with plant height was due primarily to pleiotropic effects of HD on both PH and disease. In our nursery, irrigation was applied specifically to reduce escapes due to tallness by regularly splashing inoculum up the canopy.”

The genotypic data for the underlying causal polymorphisms in the heading date genes explained the majority of the HD variation in the population in both years, as noted at various places in the results (ie. Table 2) and in the discussion on page 22 & lines 462-464: 

“The use of major genes as covariates in a SIM approach was justified as Ppd-A1, Ppd-D1 and Vrn-B1 accounted for at least two-thirds of the variation in HD and PH in 2020 and 2021, and likely played a large role in these traits in Raleigh during 2018 and 2019.”

No changes were made in response to this comment.

3. “Each genotype was planted in a single 1.3-meter row, with 3 g of seed per row except in a few cases where seed availability was lower”. “3 g of seed per 1.3-meter” that means there may -70 seeds per 1.3 meter? Is this density too high for plant growth?

Response:

That is a good estimate of the number of seeds per row, and it produces an appropriate density of wheat plants in the North Carolina environments where this experiment was conducted. No changes were made in response to this comment.

---

## [Editor Report · Decision Letter 2]

3 May 2022

Accounting for heading date gene effects allows detection of small-effect QTL associated with resistance to Septoria nodorum blotch in wheat

PONE-D-21-34614R2

Dear Dr. Rivera-Burgos,

We’re pleased to inform you that your manuscript has been judged scientifically suitable for publication and will be formally accepted for publication once it meets all outstanding technical requirements.

Kind regards,

Aimin Zhang, Ph.D.

Academic Editor

PLOS ONE
---

## [Editor Report · Acceptance letter]

10 May 2022

PONE-D-21-34614R2 

Accounting for heading date gene effects allows detection of small-effect QTL associated with resistance to Septoria nodorum blotch in wheat 

Dear Dr. Rivera-Burgos:

I'm pleased to inform you that your manuscript has been deemed suitable for publication in PLOS ONE. Congratulations! Your manuscript is now with our production department. 

Kind regards, 

on behalf of

Prof. Aimin Zhang 

Academic Editor

PLOS ONE